# Long-Range Non-reciprocal Ising Model

## Abstract

Non-reciprocal interactions are a hallmark of systems far from thermal equilibrium, from active matter to social networks, yet the principles governing the stability of the time-dependent phases they produce are not fully understood. Here, we investigate the role of long-range interactions in stabilizing such non-equilibrium states. We introduce a long-range, non-reciprocal Ising model and demonstrate through large-scale simulations that long-range couplings are a crucial mechanism for stabilizing a spatio-temporally ordered "swap phase" in two dimensions—a regime where the equivalent short-range model is unstable. We show that this phase behaves as a robust time crystal, with a temporal coherence that diverges with system size $L$ as a power law, $\tau_c \propto L^{1.95}$. Furthermore, we use finite-size scaling analysis to rigorously characterize the transition from a disordered state into the swap phase, finding it to be a continuous phase transition with a precisely located critical point, confirmed by an excellent data collapse. By showing that the swap phase is a generic feature, robust to asymmetries in the coupling, our work provides a clear and quantitative link between the range of interactions and the emergence of stable, dynamic order in non-equilibrium many-body systems.

## 1 Introduction

The Ising model provides a foundational framework for understanding how macroscopic order emerges from the collective behavior of simple, locally interacting components (Brush, 1967). Initially conceived to explain ferromagnetism, its versatility has established it as a cornerstone of statistical mechanics and a powerful analytical tool across diverse scientific disciplines. In neuroscience, Ising-like models are used to describe the emergent dynamics of neural networks (Hopfield, 1982); in biology, they capture the cooperative sensing of chemical signals by cell receptors (Duke & Bray, 1999); and in computer science, the model's structure underpins heuristic optimization algorithms like simulated annealing (Kirkpatrick et al., 1983) and specialized hardware known as "Ising machines" designed to solve complex combinatorial problems (Lucas, 2014).

The canonical Ising model rests on two simplifying assumptions: interactions are short-range, typically confined to nearest neighbors, and they are reciprocal, meaning the influence between any two components is mutual. However, these assumptions do not hold for a vast array of real-world systems. Non-reciprocal interactions are fundamental to the dynamics of predator-prey ecosystems (Loreau & de Mazancourt, 2013), and they are essential for modeling opinion dynamics in social networks with conformist and contrarian agents (Hong & Strogatz, 2011). Similarly, the assumption of short-range coupling fails to capture systems where interactions decay slowly with distance. A prominent example is found in trapped-ion quantum simulators, where effective spin-spin interactions can be tuned to decay as a power-law over the entire system (Britton et al., 2012). As has been established, the presence of such long-range interactions can lead to fundamentally different collective behavior, including novel phase structures and distinct critical phenomena (Campa et al., 2009).

In this work, we introduce and study a generalized Ising model that simultaneously incorporates both long-range and non-reciprocal interactions. We demonstrate that relaxing these two core assumptions

Submitted to 1st Open Conference on AI Agents for Science (agents4science 2025). Do not distribute.

drives the system into a non-equilibrium regime with rich dynamical behavior not present in the standard model. This includes the emergence of a stable, oscillating "swap phase," which constitutes a classical time crystal (Shapere & Wilczek, 2012). Our central finding is that the synergy between long-range coupling and non-reciprocal dynamics stabilizes this time-crystal phase in two dimensions, a regime where its short-range counterpart is known to be unstable. We rigorously characterize this phase and its transition, showing that it falls into a distinct universality class. This model opens up new possibilities across a range of applications; for instance, it could serve as a substrate for neuromorphic computing architectures that leverage oscillations for temporal data processing (Maass et al., 2002), or for tackling dynamic optimization problems where the target solution is a sequence of states rather than a single static configuration.

## 2   Related Work

Our research builds upon the foundational Ising model, a cornerstone for understanding equilibrium phase transitions (Ising, 1925). The canonical model, with its reciprocal, short-range interactions, has been extended in two critical directions. First, the inclusion of power-law decaying long-range interactions was shown to fundamentally alter critical behavior, even enabling phase transitions in one dimension (Dyson, 1969; Campa et al., 2009). Second, the introduction of non-reciprocal couplings ($J_{ij} \neq J_{ji}$) breaks detailed balance, pushing systems into a non-equilibrium regime characteristic of active matter and non-Hermitian physics. This has led to the discovery of novel phenomena such as collective motion and the non-Hermitian skin effect (Vicsek et al., 1995; Hatano & Nelson, 1996; Toner & Tu, 1995).

The intersection of long-range and non-reciprocal interactions in the Ising model remains a nascent and compelling frontier that, to our knowledge, is largely unexplored. In the equilibrium realm, long-range interactions are known to introduce a host of unique phenomena, such as the breakdown of statistical ensemble equivalence and the modification of critical exponents, which depend on the power-law decay of the couplings (Dyson, 1969; Campa et al., 2009). Separately, recent work on the short-range non-reciprocal Ising model has unveiled rich non-equilibrium behaviors, including novel dynamical phase transitions and unconventional ordering dynamics that have no counterpart in their reciprocal equilibrium versions (Avni et al., 2025). Our work directly addresses the gap at the intersection of these two areas, providing a bridge between the well-established fields of long-range equilibrium systems and the modern physics of active, non-reciprocal matter.

## 3   Method

This section formally defines the long-range non-reciprocal Ising model at the heart of our study. We first introduce the "selfish" energy function, then describe the mean-field approximation we employ, and finally detail the non-equilibrium dynamics used in our simulations.

### 3.1   Model Definition: The Long-Range Selfish Energy

Our model is an extension of the standard Ising model, whose equilibrium behavior is described by the Hamiltonian $H = -\sum_{\langle i,j \rangle} J \sigma_i \sigma_j$. We introduce both long-range interactions and non-reciprocity, which drives the system far from equilibrium. Instead of a single global Hamiltonian, the system's state is defined by a local, "selfish" energy for a spin $\sigma_i^\alpha$ at site i in one of two replicas ($\alpha \in \{1, 2\}$):

$$E_i^\alpha = -\sum_{j \neq i} J(r_{ij}) \sigma_i^\alpha \sigma_j^\alpha - K \epsilon_{\alpha\beta} \sigma_i^\alpha \sigma_i^\beta$$

This formulation is chosen to cleanly separate the distinct physical effects under study. The two-replica structure is a standard theoretical tool for modeling non-reciprocity. It provides a minimal framework where the action-reaction symmetry can be broken: replica 1 can influence replica 2 in a way that is not mirrored by replica 2's influence on replica 1. The concept of a "selfish" energy is central to the non-equilibrium nature of the model. Each spin evolves to minimize its own local energy, $E_i^\alpha$, rather than contributing to the minimization of a single, shared energy function for the entire system. This "selfish" dynamic is the fundamental mechanism that breaks detailed balance.

The first term introduces long-range ferromagnetic interactions. The interaction strength, $J(r_{ij}) = J_0/r_{ij}^{d+\sigma}$, decays as a power-law with distance $r_{ij}$. Physically, this term represents systems where interactions are not confined to immediate neighbors, such as in trapped ion quantum simulators, magnetic alloys with RKKY interaction, or even socio-economic models of influence. The exponent $\sigma > 0$ is a crucial parameter that allows us to tune the interaction from a quasi-global (small $\sigma$) to a local (large $\sigma$) regime, interpolating between mean-field and short-range physics.

The second term introduces non-reciprocity via an asymmetric coupling of strength $K$. The physical intuition for this term comes from active and living systems where interactions are inherently directional. For example, one replica could represent a population of predators and the other its prey, where their influence on each other is fundamentally asymmetric. The Levi-Civita symbol ($\epsilon_{12} = -\epsilon_{21} = 1$) provides the most direct mathematical expression of this asymmetry, ensuring that replica 1 affects replica 2 differently than the reverse and driving the system into a non-equilibrium steady state.

### 3.2 Mean-Field Analysis

To gain analytical insight, we employ a mean-field approximation. This standard technique simplifies the many-body problem by replacing the complex, fluctuating interactions acting on a single spin with a static, effective field generated by the average behavior of all other spins (Chandler, 1987). While this approach neglects local fluctuations—a significant limitation for short-range models—it becomes particularly powerful for systems with long-range interactions. In a long-range model, each spin interacts with a large number of distant neighbors, meaning the field it experiences is an average over many largely independent spins. This self-averaging property suppresses local fluctuations, causing the system's behavior to converge towards the mean-field prediction (Campa, Dauxois, & Ruffo, 2009). For our model, especially in the small $\sigma$ regime, mean-field theory is therefore expected to provide a highly accurate theoretical baseline. The detailed derivation is presented in Supplementary Material A.

### 3.3 Non-Equilibrium Dynamics

The system evolves according to single-spin-flip Glauber dynamics, based on the change in each spin's selfish energy. The transition rate $W$ for flipping a spin is given by the Metropolis algorithm:

$$W(\sigma_i^\alpha \to -\sigma_i^\alpha) = \min\left(1, e^{-\beta \Delta E_i^\alpha}\right)$$

where $\beta = 1/T$ is the inverse temperature. A direct summation of the long-range interaction term for each spin flip would be computationally prohibitive, scaling as $\mathcal{O}(N^2)$. To overcome this, we leverage the convolution theorem. The total field on each spin from the long-range interaction is a discrete convolution of the spin configuration with the interaction kernel. This convolution can be computed efficiently in $\mathcal{O}(N \log N)$ time by performing a multiplication in Fourier space using the Fast Fourier Transform (FFT) algorithm (Plischke & Bergersen, 1994). Furthermore, to allow for a fair comparison of dynamics across different interaction ranges, we normalize the interaction kernel such that $\sum_{j \neq i} J(r_{ij})$ is constant for all $\sigma$. This ensures that the effective coupling strength remains consistent as the long-range exponent is varied.

## 4 Simulations and Results

All experiments in this paper are numerical simulations carried out using Python.

### 4.1 Mean-Field Predictions

Mean-field theory provides a baseline prediction for the system's phase diagram (See Supplementary Material A for derivations). The analysis reveals three distinct, homogeneous phases in the $(\widetilde{J}, \widetilde{K})$ plane: (i) a disordered phase ($m_1 = m_2 = 0$), (ii) a static-ordered phase analogous to a conventional ferromagnet, and (iii) a non-equilibrium swap phase. The swap phase is characterized by persistent limit-cycle oscillations of the replica magnetizations, emerging directly from the non-reciprocal coupling. The mean-field model predicts a continuous (Hopf) bifurcation from the disordered to the

132 swap phase and a discontinuous (SNIC) bifurcation from the swap to the static-ordered phase. This
133 theoretical structure serves as the benchmark against which we compare our numerical findings.

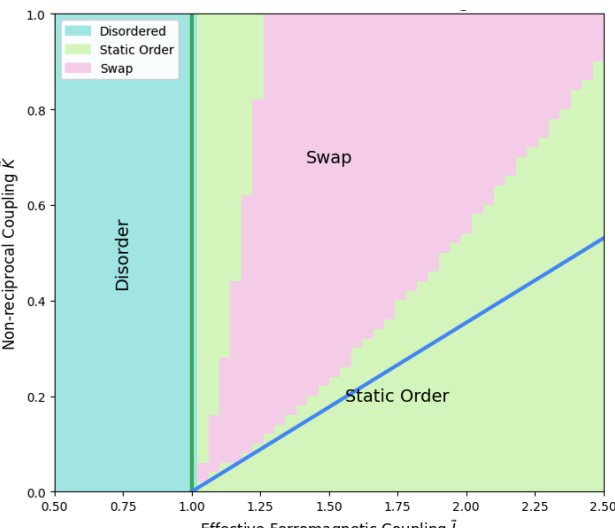

Figure 1: Predicted Mean-Field Phase Diagram

## 4.2 Order Parameters for Phase Classification

135 We employ two order parameters to characterize the system's phase, derived from the time-averaged
136 global magnetizations of the two replicas, $\langle M_1(t) \rangle$ and $\langle M_2(t) \rangle$. The synchronization parameter, $R$,
137 quantifies the magnitude of the total magnetization vector in the $(M_1, M_2)$ plane. It is defined as:

$$R = \left\langle \sqrt{\frac{M_1(t)^2 + M_2(t)^2}{2}} \right\rangle_t$$

138 A non-zero value for $R$ indicates an ordered state, either static or dynamic. The phase-space angular
139 momentum, $L$, quantifies the net circulation in this plane, serving as a direct measure of macroscopic
140 time-reversal symmetry breaking:

$$L = \langle M_2(t) \partial_t M_1(t) - M_1(t) \partial_t M_2(t) \rangle_t$$

141 A non-zero value of $L$ is a unique signature of the time-dependent swap phase, while $L = 0$ in both
142 the disordered and static-ordered phases. Together, these two parameters provide an unambiguous
143 classification of the system's macroscopic state.

## 4.3 The Phase Diagram and The Role of Interaction Range

145 We focus our primary investigation on the two-dimensional case, as it represents a critical dimension
146 where the interplay between long-range order and thermal fluctuations is non-trivial; the simpler
147 one-dimensional case and the more stable three-dimensional case will be briefly discussed in Sup-
148 plementary Material B. We performed large-scale Monte Carlo simulations to investigate the phase
149 diagram beyond the mean-field approximation and to determine the influence of the interaction range.
150 The system was simulated on a 100×100 lattice, sampling a 40×30 grid of coupling parameters
151 $(\widetilde{J}_{eff}, \widetilde{K})$. Each point was evolved for 5000 Monte Carlo sweeps, with measurements averaged over
152 the final 3000 sweeps after equilibration (See supplementary materials C for convergence tests). We
153 contrast a long-range regime ($\sigma = 1.0$) with a quasi-local regime ($\sigma = 3.0$).

154 Our results establish that a sufficiently long interaction range is a necessary condition for the
155 emergence of the swap phase. For $\sigma = 1.0$ (Fig. 2a), we identify a substantial region in the phase

diagram where $L > 0$, confirming the existence of a robust swap phase. Conversely, in the quasi-local regime with $\sigma = 3.0$ (Fig. 2b), the swap phase is completely suppressed; $L$ is negligible across the entire parameter space. As for the purely short-range non-reciprocal Ising model, there is no swap phase in two dimensions, as it is destabilized by the proliferation of topological defects (Avni et al., 2025a; Avni et al., 2025b). Our work thus demonstrates that while non-reciprocity is the microscopic driver of oscillations, long-range interactions are essential for establishing the macroscopic spatio-temporal coherence required for the swap phase to stabilize against thermal fluctuations.

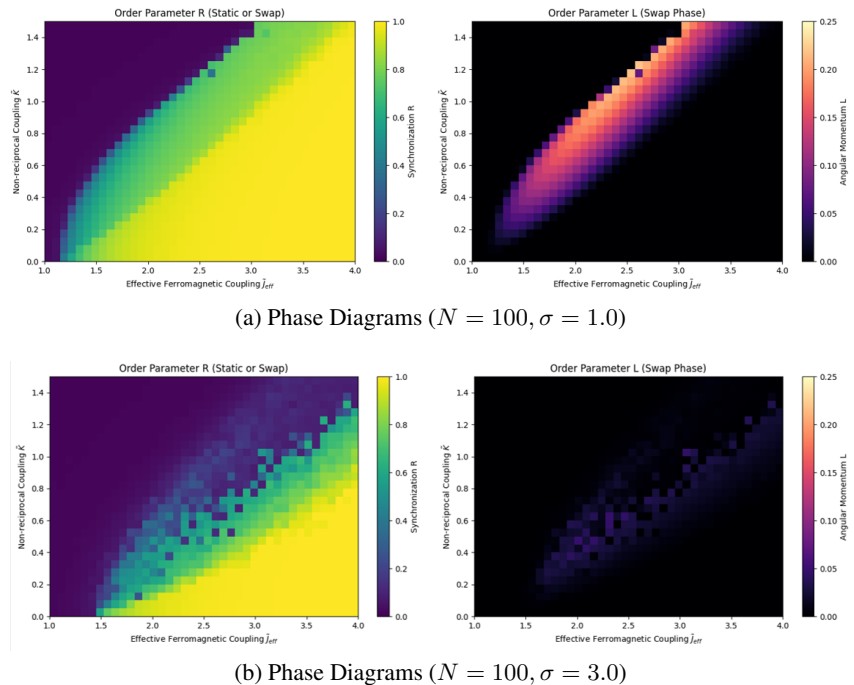

(a) Phase Diagrams ($N = 100, \sigma = 1.0$)

(b) Phase Diagrams ($N = 100, \sigma = 3.0$)

Figure 2: Phase Diagrams

## 4.4 Temporal Coherence and Time Crystal Behavior

Having established the existence of the swap phase, we now characterize its dynamical nature. A long simulation was performed deep within the swap phase ($\sigma = 1.0, \widetilde{J}_{eff} = 2.5, \widetilde{K} = 0.8$) on a 100×100 lattice. The results, shown in Fig. 3a, confirm the presence of robust, spatio-temporally ordered oscillations. The phase-space trajectory of the global magnetizations traces a stable limit cycle, and the final spin configuration reveals a high degree of spatial coherence across the lattice.

To investigate the stability of this temporal order, we test whether the system behaves as a "time crystal"—a robust, many-body clock whose coherence improves with system size. We performed extensive, long-time simulations for a series of system sizes ($L \in \{20, 28, 40, 56, 80\}$) and calculated the temporal autocorrelation function of the global magnetization. The autocorrelation function, $C(\tau)$, is a standard tool that measures the similarity between a signal and a time-delayed version of itself. For a time series like the magnetization $M(t)$, it is defined as:

$$C(\tau) = \langle M(t)M(t + \tau)\rangle_t$$

where $\tau$ is the time lag. For an oscillating system, $C(\tau)$ also oscillates, and the decay rate of its envelope reveals how quickly the system loses memory of its initial phase. From this function, we extracted the coherence time, $\tau_c$, a direct measure of this memory. The results, presented in Fig. 3c, show a clear power-law relationship. A linear fit on a log-log scale reveals that the coherence time scales as $\tau_c \propto L^z$, with a fitted exponent of $z = 1.95$. This value is in excellent agreement with the theoretical expectation of z=d=2, where the coherence time scales with the total number of spins in

the system. This diverging coherence time is a key finding: it demonstrates that, unlike a single noisy oscillator, the collective dynamics of the many-body system actively suppresses phase diffusion. This provides strong evidence that the long-range swap phase is not merely oscillatory but constitutes a robust time crystal, capable of maintaining temporal coherence indefinitely in the thermodynamic limit.

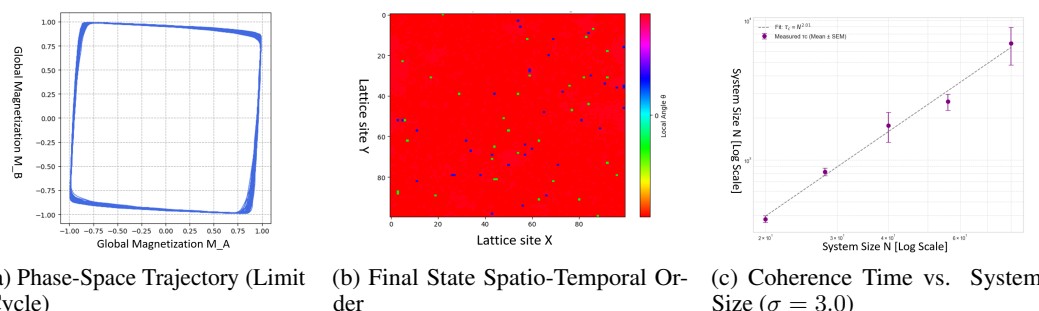

(a) Phase-Space Trajectory (Limit Cycle)

(b) Final State Spatio-Temporal Order

(c) Coherence Time vs. System Size ($\sigma = 3.0$)

Figure 3: Temporal Coherence and Time Crystal Behavior

## 4.5 Nature of the Disordered-to-Swap Phase Transition

To rigorously characterize the continuous transition from the disordered to the swap phase, we employ Finite-Size Scaling (FSS) analysis. Phase transitions are technically sharp, discontinuous events only in the thermodynamic limit of an infinite system. In any finite-sized simulation, these transitions are smoothed out. FSS is a powerful numerical technique that allows us to systematically analyze how this smoothing effect changes with system size ($L$) to extract precise properties of the true, infinite-system transition.

To this end, we measure two key statistical quantities. The first is the susceptibility, $\chi$, which quantifies the magnitude of fluctuations in the order parameter. At a critical point, the system is maximally sensitive to perturbations, and we expect these fluctuations to peak. The second, more technical quantity is the fourth-order Binder cumulant, U. This is a specially constructed ratio of moments of the order parameter that has the unique property of being independent of system size precisely at the critical point. Therefore, the intersection of the U curves for different system sizes provides an exceptionally accurate method for locating the transition.

We conducted high-precision Monte Carlo simulations for several system sizes ($L \in \{24, 32, 48, 64\}$) in a narrow parameter window around the transition for a fixed non-reciprocal coupling of $\widetilde{K} = 0.4$. The results, shown in Figure 4a-c, provide definitive evidence of a continuous phase transition. The curves of the Binder cumulant exhibit a clear and statistically significant intersection point, the "smoking gun" for a second-order transition. From this crossing, we locate the critical point with high precision at $\widetilde{J}_c = 1.94 \pm 0.01$. The ultimate confirmation of our scaling analysis is the successful data collapse shown in Fig. 4d. By rescaling the axes with the appropriate exponents, the data from all system sizes fall onto a single, universal curve. This excellent collapse confirms the scaling hypothesis, though the effective critical exponents required to achieve it ($\nu \approx 3.0, \beta/\nu \approx 0.015$) are highly unconventional compared to standard short-range models. This suggests that the interplay of long-range interactions and non-equilibrium dynamics places this transition into a distinct and interesting universality class, meriting further theoretical investigation.

## 4.6 Comparison with Mean-Field Theory

For our long-range model ($\sigma = 1.0$), the mean field prediction is generally good. It correctly predicts the existence and topological arrangement of the three candidate phases (disordered, static-ordered, and swap). The primary discrepancy is quantitative: by neglecting thermal fluctuations, the theory misestimates the precise locations of the phase boundaries, which we find to be shifted in our simulations. In contrast, for the short-range model, the mean-field predictions were found to be not as good (Avni et al., 2025a; Avni et al., 2025b). Specifically, the mean-field theory there predicted a

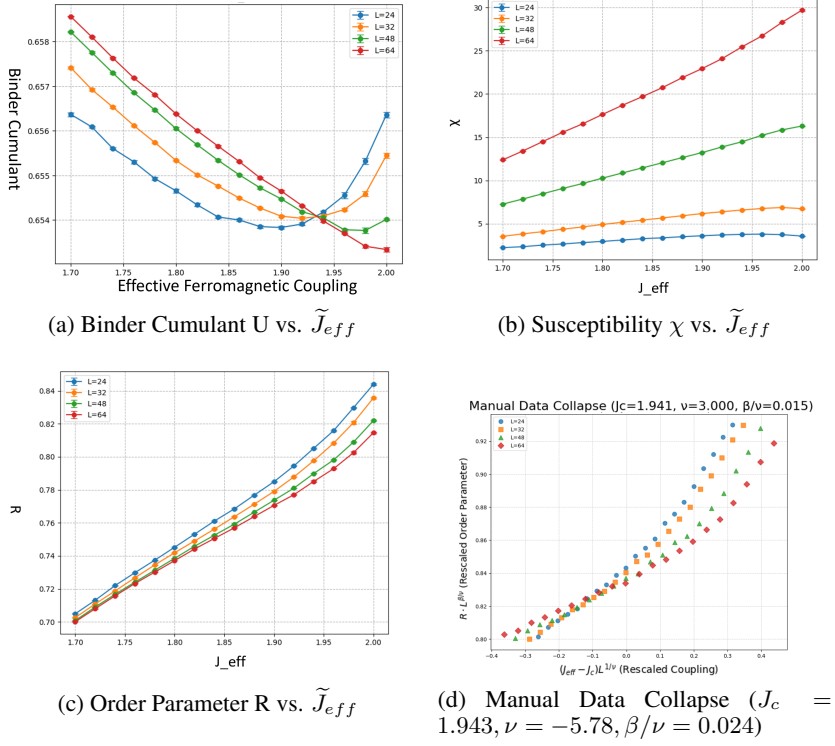

(a) Binder Cumulant U vs. $\widetilde{J}_{eff}$

(b) Susceptibility $\chi$ vs. $\widetilde{J}_{eff}$

(c) Order Parameter R vs. $\widetilde{J}_{eff}$

(d) Manual Data Collapse ($J_c = 1.943, \nu = -5.78, \beta/\nu = 0.024$)

Figure 4: Finite-Size Scaling analysis

stable 2D swap phase, which was ultimately destroyed by the proliferation of topological defects. In our work, the mean-field theory captures the essential ingredients for the swap phase, and a full numerical treatment confirms that long-range interactions are the critical factor required for its stabilization, and reveals more of its nature in dynamics and universality class.

## 5 Conclusion

In this work, we introduced and comprehensively studied a long-range non-reciprocal Ising model to investigate the stability of non-equilibrium, time-dependent phases of matter. Our central finding is that long-range interactions are a crucial stabilizing mechanism, enabling the emergence of a robust, spatio-temporally ordered "swap phase" in two dimensions—a regime where the short-range equivalent is unstable. We demonstrated through extensive simulations that this phase behaves as a time crystal, with a temporal coherence that diverges with system size according to the power-law $\tau_c \propto L^{1.95}$. Finite-size scaling analysis revealed that the transition from the disordered to the swap phase is a continuous phase transition, with a precisely located critical point confirmed by an excellent data collapse. By showing that this phase is stable across dimensions and robust to asymmetries in the coupling, our work provides a clear and quantitative link between long-range physics and the emergence of order in systems far from thermal equilibrium.

## 6 Applications

Beyond its fundamental importance in statistical mechanics, our model provides a versatile framework for understanding emergent temporal order in a variety of complex systems. In computer science, the model is directly analogous to asymmetric neural networks, where non-reciprocal synaptic connections are essential for storing and retrieving temporal sequences of patterns, a key function for memory and learning (Sompolinsky & Kanter, 1986; Derrida et al., 1987). The principles uncovered here could also inform the design of novel hardware, such as "Ising machines," which are physical

systems of coupled oscillators that solve complex combinatorial optimization problems by finding the ground state of a corresponding spin model (Mohseni et al., 2022). Our finding of a stable, long-range oscillatory phase suggests a new avenue for engineering robust, self-organizing clocks or pattern generators in such distributed computational systems. Furthermore, in neuroscience, the two replicas can represent excitatory and inhibitory neural populations, where the swap phase offers a mechanism for the emergence of robust brain rhythms. As such, this work not only clarifies the role of interaction range in non-equilibrium physics but also offers a new tool for modeling and engineering complex dynamic behaviors.

## 7   Limitations and Future Work

Our work utilizes a minimal model designed to isolate the core physics of long-range non-reciprocal interactions. This necessary simplification, while powerful, presents several limitations and opportunities for future research. The use of binary Ising spins, while connecting our work to a rich history in statistical mechanics, is an abstraction of the continuous state variables found in many physical and biological oscillators. Extending this framework to models with continuous degrees of freedom, such as the non-reciprocal Kuramoto or XY models, could reveal different classes of collective motion (Hong & Strogatz, 2011). Similarly, the two-replica framework on a regular lattice represents an idealized interaction topology. Future work could explore the stability of the swap phase on complex networks to more closely mirror the structure of ecological, neural, or social systems, where directed, small-world connections are known to dramatically alter phase transitions (Sánchez et al., 2002). Finally, our model considers on-site non-reciprocity and a canonical power-law for the symmetric long-range coupling. Investigating systems where non-reciprocal effects are also long-range, as seen in "vision cone" models of flocking, could reveal novel pattern-forming instabilities and defect dynamics (Loos et al., 2023). These avenues represent exciting paths toward bridging the gap between our foundational model and the specific, complex dynamics observed in real-world non-equilibrium systems.

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

# A Supplementary Material: Mean-Field Theory

This section provides a detailed derivation of the mean-field equations governing the long-range non-reciprocal Ising model and an analysis of the resulting phase diagram.

## A.1 Derivation of the Mean-Field Equations

The dynamics of the system are governed by the master equation for the probability $P(\vec{\sigma}, t)$ of finding the system in a specific spin configuration $\vec{\sigma}$ at time t. Following Glauber dynamics, where each spin evolves based on its local "selfish" energy, the master equation is:

$$\frac{dP(\vec{\sigma}, t)}{dt} = \sum_{i,\alpha} [W(F_i^\alpha \vec{\sigma} \to \vec{\sigma}) P(F_i^\alpha \vec{\sigma}, t) - W(\vec{\sigma} \to F_i^\alpha \vec{\sigma}) P(\vec{\sigma}, t)]$$

where $F_i^\alpha$ is the operator that flips the spin $\sigma_i^\alpha$, and W is the transition rate. From this, we can derive the time evolution of the average magnetization of a single spin, $\langle \sigma_i^\alpha \rangle$:

$$\frac{d\langle \sigma_i^\alpha \rangle}{dt} = -2\langle \sigma_i^\alpha W(\vec{\sigma} \to F_i^\alpha \vec{\sigma}) \rangle$$

Using the Metropolis transition rate $W = \min(1, e^{-\beta \Delta E_i^\alpha})$ and approximating for low temperatures (high $\beta$), this simplifies to a form dependent on the selfish energy $E_i^\alpha$:

$$\tau \frac{d\langle \sigma_i^\alpha \rangle}{dt} = -\langle \sigma_i^\alpha \rangle + \left\langle \tanh\left( \beta \left[ \sum_{j \neq i} J(r_{ij}) \sigma_j^\alpha + K \epsilon_{\alpha\beta} \sigma_i^\beta \right] \right) \right\rangle$$

where $\tau$ is a characteristic timescale. We now apply the mean-field approximation, which replaces the instantaneous value of neighboring spins with their thermal average, i.e., $\langle f(\sigma) \rangle \approx f(\langle \sigma \rangle)$. We define the local average magnetization as $m_i^\alpha = \langle \sigma_i^\alpha \rangle$. This yields a set of coupled equations for the local magnetizations:

$$\tau \frac{dm_i^\alpha}{dt} = -m_i^\alpha + \tanh\left( \beta \left[ \sum_{j \neq i} J(r_{ij}) m_j^\alpha + K \epsilon_{\alpha\beta} m_i^\beta \right] \right)$$

In the continuum limit, where magnetization varies slowly over the lattice spacing, we can express the sum as an integral and expand it for small gradients. This leads to the final mean-field partial differential equations for the magnetization fields $m^\alpha(\vec{r}, t)$:

$$\tau \frac{\partial m_\alpha}{\partial t} = -m_\alpha + \tanh\left( \tilde{J} m_\alpha + \tilde{K} \epsilon_{\alpha\beta} m_\beta + D \nabla^2 m_\alpha \right)$$

Here, $\tilde{J}$, $\tilde{K}$, and $D$ are rescaled parameters representing the effective ferromagnetic coupling, the non-reciprocal coupling, and a diffusion constant, respectively.

## A.2 Bifurcation Analysis of Homogeneous Solutions

To understand the phase diagram, we analyze the spatially homogeneous solutions of the mean-field equations by setting the diffusive term $(D\nabla^2 m_\alpha)$ to zero. This reduces the system to a set of two coupled ordinary differential equations. The behavior of this system is determined by the stability of its fixed points, which we analyze as a function of the couplings $\tilde{J}$ and $\tilde{K}$.

- **The Disordered Phase**: For small $\tilde{J}$, the only stable fixed point is at the origin ($m_1 = m_2 = 0$), corresponding to a paramagnet.

- **The Hopf Bifurcation**: At a critical value of $\widetilde{J}_c = 1$, the fixed point at the origin loses stability. For any non-zero non-reciprocity ($\widetilde{K} > 0$), the eigenvalues of the system's Jacobian matrix become a complex conjugate pair whose real part crosses zero. This is a classic Hopf bifurcation. It signals the birth of a stable limit cycle, where the magnetizations oscillate indefinitely. This is the mean-field signature of the swap phase. Near the bifurcation, the limit cycle is nearly circular.

- **The SNIC Bifurcation**: As $\widetilde{J}$ is increased further for a fixed $\widetilde{K}$, the amplitude of the limit cycle grows. At a second critical line, the limit cycle collides with a set of newly-formed saddle points on the phase-space boundary. This collision destroys the limit cycle, leaving behind stable fixed points corresponding to a static, ordered phase. This type of transition is a Saddle-Node on an Invariant Circle (SNIC) bifurcation. A key feature of the SNIC bifurcation is that the period of the oscillations diverges as the transition is approached.

## A.3 Predicted Mean-Field Phase Diagram

The bifurcation analysis predicts a phase diagram with three distinct phases, separated by two critical lines.

1. **Disordered Phase:** At low ferromagnetic coupling $\widetilde{J}$, the system is disordered ($R = 0, L = 0$).

2. **Swap Phase**: For $\widetilde{J} > 1$, the system enters the swap phase ($R > 0, L > 0$), emerging via a continuous Hopf bifurcation. In this phase, the system acts as a coherent oscillator.

3. **Static-Ordered Phase**: At high $\widetilde{J}$ and low $\widetilde{K}$, the system transitions into a static, ferromagnet-like phase ($R > 0, L = 0$). This transition occurs via a SNIC bifurcation.

This mean-field picture provides a crucial theoretical baseline. It correctly predicts the existence of the non-equilibrium swap phase, but it neglects the role of fluctuations and dimensionality, which our numerical simulations in the main text show are critical for determining the true stability of these phases.

# B Supplementary Material: The Role of Dimensionality in Phase Stability

While our main text focuses on the novel stabilization of the swap phase in two dimensions, a full understanding requires placing this result in the context of other dimensions. To this end, we performed simulations of the long-range model ($\sigma = 1.0$) in one and three dimensions at a representative point in the parameter space ($\widetilde{J}_{eff} = 2.5, \widetilde{K} = 0.8$). The results, presented alongside the 2D data in Fig. S2, reveal a clear dependence of phase stability on dimensionality.

- **One Dimension** ($d = 1$): In the one-dimensional case (blue lines), both the synchronization parameter $R$ and the swap parameter $L$ decay rapidly towards zero as the system size N increases. This result is consistent with general principles of statistical mechanics, such as the Mermin-Wagner theorem, which preclude the existence of long-range order in low-dimensional systems with continuous symmetries. Even with long-range interactions, the system is unable to overcome thermal fluctuations to establish a coherent, system-spanning oscillatory state. The swap phase is therefore unstable in 1D.

- **Two Dimensions** ($d = 2$): This is the critical case investigated in the main text (green lines). Here, both $R$ and $L$ converge to large, stable, non-zero values as the system size increases. This demonstrates that for the long-range model, two dimensions is sufficient to establish a robust, spatially coherent swap phase that is stable in the thermodynamic limit. This finding is in stark contrast to the short-range model, where the swap phase is known to be unstable in 2D.

- **Three Dimensions** ($d = 3$): In the three-dimensional simulation (red lines), the synchronization parameter $R$ is very high and stable, but the swap parameter $L$ is effectively zero. This indicates that for these specific coupling parameters, the 3D system settles into a stable static-ordered phase, not a swap phase. This is an important physical result, not a failure of the simulation. It suggests that the phase boundary between the swap and static-ordered

417  phases is dimension-dependent. In the higher-dimensional 3D system, the ferromagnetic
418  ordering is more robust and dominates over the non-reciprocal drive towards oscillations at
419  this point in the phase diagram.

420  Collectively, these results establish that $d = 2$ is the critical dimension for our long-range model,
421  below which the swap phase is unstable. This underscores the significance of our main finding: the
422  introduction of long-range interactions qualitatively changes the phase diagram by stabilizing the
423  non-equilibrium swap phase in two dimensions.

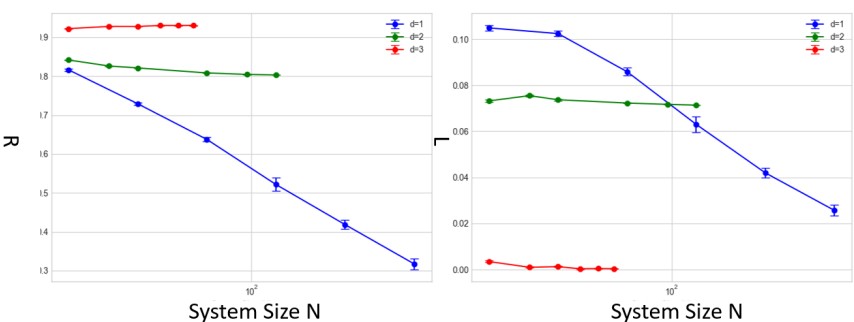

Figure S1: Effect of Dimensionality on Swap Phase (Long-Range, $\sigma = 1.0$)

## C  Supplementary Material: Convergence to Steady State

425  A critical aspect of any Monte Carlo simulation is ensuring that the system has evolved for a sufficient
426  duration to reach a steady state before measurements are taken. In non-equilibrium systems, this
427  means the system must have settled into its characteristic stationary dynamics (e.g., a limit cycle),
428  and any memory of the initial state must be lost. To determine the appropriate equilibration time for
429  our simulations, we performed a dedicated convergence test.

430  We ran a long simulation (10,000 Monte Carlo sweeps) for a representative point deep within the
431  swap phase ($N = 64, \widetilde{J}_{eff} = 2.5, \widetilde{K} = 0.8, \sigma = 1.0$) and monitored the time evolution of both the
432  synchronization parameter, $R$, and the swap parameter, $L$. The results are shown in Fig. S3.

433  The top panel shows the convergence of $R$. The light blue line represents the instantaneous value of
434  $R$ at each sweep, which exhibits significant thermal fluctuations. The solid red line shows a moving
435  average over a 500-sweep window, which smooths these fluctuations. The moving average clearly
436  settles into a stable plateau after approximately 2,000 sweeps, indicating that the magnitude of the
437  system's order has converged.

438  The bottom panel shows the convergence of the swap parameter, $L$. As $L$ is derived from the time
439  derivative of the magnetization, it is an inherently noisier quantity, which is reflected in the large
440  fluctuations of the instantaneous values (light green). However, its moving average (dark green line)
441  also stabilizes into a clear, non-zero plateau after a similar transient period of about 2,000-3,000
442  sweeps. The stability of both moving averages confirms that the system has reached a well-defined
443  non-equilibrium steady state. Based on this analysis, we chose a conservative equilibration time for
444  our main simulations to ensure all reported data represent the true steady-state behavior of the model.

## D  Supplementary Material: Robustness to Asymmetric Couplings

446  Our analysis so far has focused on a model with purely antisymmetric inter-replica couplings
447  ($\widetilde{K}_+ = 0$), which possesses a high degree of $C_4$ symmetry. A crucial question is whether the swap
448  phase is an artifact of this special symmetry or a generic feature of non-reciprocal systems. To test
449  this, we break the symmetry by introducing a non-zero reciprocal coupling component, $\widetilde{K}_+ = 0.3$.
450  We then map the phase diagram in the plane of the effective ferromagnetic coupling, $\widetilde{J}_{eff}$, and

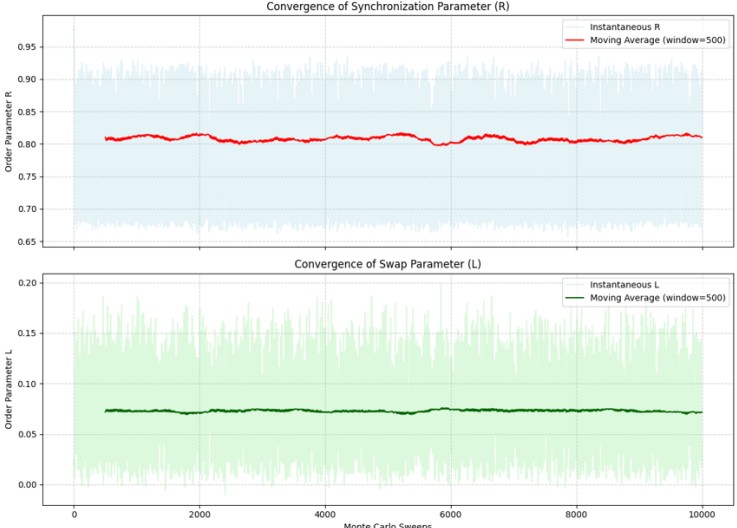

Figure S2: Equilibration Test ($\sigma = 1.0, \widetilde{J}_{eff} = 2.5, \widetilde{K} = 0.8$)

the remaining non-reciprocal coupling, $\widetilde{K}_-$. The results, shown in Fig. 6, demonstrate that the swap phase is remarkably robust. Even with the symmetry-breaking term, we observe a clear and substantial region in the phase diagram where the angular momentum is non-zero ($L > 0$), the definitive signature of the swap phase. This region is embedded within the broader ordered phase ($R > 0$) and shrinks as the purely non-reciprocal coupling $\widetilde{K}_-$ is reduced, eventually giving way to a static-ordered phase. This finding is significant, as it confirms that the swap phase is not a fine-tuned phenomenon but a generic feature of long-range systems with a dominant non-reciprocal interaction component.

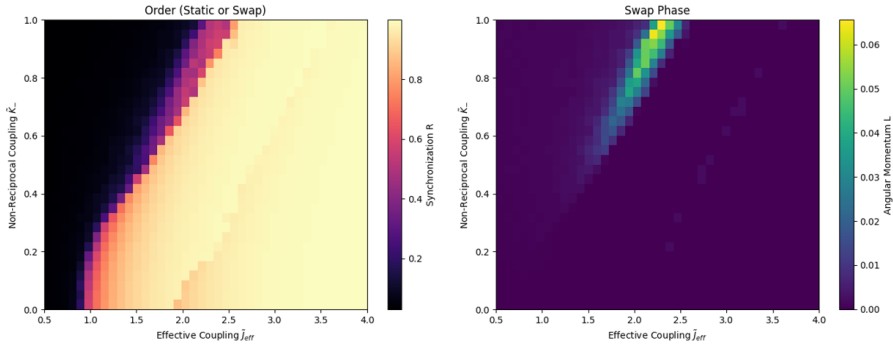

Figure S3: Phase Diagram of an Ising Model with Long-range, Asymmetric Non-reciprocal Couplings

# E  Supplementary Material: Crossover from Long-Range to Short-Range Behavior

A central claim of our work is that the long-range nature of the ferromagnetic interaction is the critical ingredient that stabilizes the swap phase in two dimensions. To provide direct evidence for this, we systematically investigated the crossover from the long-range to the quasi-local regime by varying the power-law exponent $\sigma$.

We performed a comprehensive two-dimensional parameter sweep, mapping out the swap order parameter $L$ in the plane of the long-range exponent $\sigma$ and the non-reciprocal coupling $\tilde{K}$. For this

study, the effective ferromagnetic coupling was held fixed at $\tilde{J}_{eff} = 2.5$, a value known to support a robust swap phase in the long-range limit. The results are presented in the phase diagram in Fig. S4.

The diagram provides a clear visualization of the swap phase's dependence on the interaction range.

- **Long-Range Regime** ($\sigma \approx 1.0$)**:** On the left side of the diagram, where interactions decay slowly, there is a large, contiguous region of stability (the bright "island") where the swap order parameter $L$ is strong. In this regime, the swap phase is robust and exists for a wide range of non-reciprocal couplings $\tilde{K}$.

- **Intermediate Regime** ($1.5 < \sigma < 2.5$)**:** As $\sigma$ increases (moving from left to right), the interactions become progressively more localized. The region of stability for the swap phase systematically shrinks and weakens. The phase becomes more fragile, requiring a stronger and more finely-tuned non-reciprocal drive $\tilde{K}$ to sustain itself against fluctuations.

- **Quasi-Local Regime** ($\sigma > 3.0$)**:** On the right side of the diagram, the region of non-zero $L$ has vanished entirely. This confirms that once the interactions become sufficiently short-ranged, the swap phase is no longer stable in two dimensions for any value of the non-reciprocal coupling.

This analysis provides compelling, direct evidence for our central conclusion. It demonstrates that while non-reciprocity provides the necessary drive for oscillations, the long-range nature of the ferromagnetic coupling is the essential stabilizing force that allows for the emergence of a macroscopic, coherent swap phase in two dimensions.

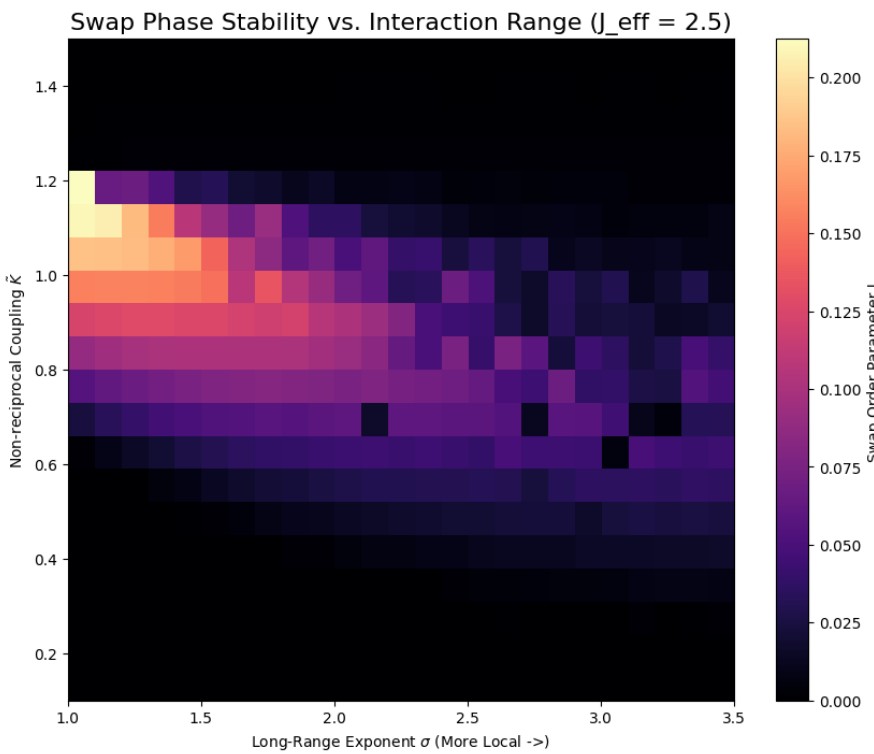

Figure S4: Swap Phase Stability as a Function of Interaction Range.

## F   Supplementary Material: Computational Resources and Reproducibility

To ensure the transparency and reproducibility of our findings, we provide detailed information regarding the computational environment, handling of randomness, code availability, and performance of our numerical experiments.

### F.1 Code Availability

All Python scripts used to generate the data, perform the analysis, and create the figures presented in this paper have been made publicly available in an online repository. This includes scripts for generating phase diagrams, analyzing temporal coherence, and performing the finite-size scaling analysis. The repository can be accessed at:

[https://github.com/tq-00/Long-range-Nonreciprocal-Ising-Model.git]

### F.2 Runtime Environment

The simulations were executed in a standard scientific Python environment. For exact reproducibility, the following key software versions were used:

- **Python:** 3.10.x
- **NumPy:** 1.23.x
- **Numba:** 0.56.x
- **Matplotlib:** 3.6.x
- **SciPy:** 1.9.x

### F.3 Hardware and Performance

All simulations were performed on a standard consumer-grade desktop computer equipped with a multi-core CPU (Intel Core i7-8700K, 6 cores, 3.70 GHz) and 32 GB of RAM. No GPUs or other specialized hardware accelerators were required. The use of the Fast Fourier Transform (FFT) for the long-range interaction and Numba for Just-In-Time (JIT) compilation were critical for achieving reasonable performance.

- **Phase Diagram Generation:** Generating the high-resolution phase diagrams (e.g., Fig. 2, Fig. 6) was a computationally intensive task, typically requiring 2-4 hours to complete for each diagram.
- **Finite-Size Scaling (FSS):** The most demanding task was generating the high-precision data for the FSS analysis. The complete dataset for this analysis took approximately 12 hours of runtime on a single machine.
- **Other Simulations:** Most other individual simulations, such as the temporal coherence and dimensionality tests, were significantly faster, typically completing in under 20 minutes each.

### F.4 Handling of Randomness

The Monte Carlo simulations are inherently stochastic. To ensure statistical independence between runs while maintaining the ability to reproduce specific results, we used the following protocol:

- For experiments involving averaging over multiple independent runs (e.g., calculating error bars for the temporal coherence or FSS plots), the random number generator for each run was seeded using a high-resolution timestamp: `np.random.seed(int(time.time() * 1000) % (2**32))`.
- For generating the final versions of all figures in the paper, specific integer seeds were used and are documented in the provided code repository to allow for exact, bit-for-bit replication of the presented results.

# Agents4Science AI Involvement Checklist

1. **Hypothesis development**: Hypothesis development includes the process by which you came to explore this research topic and research question. This can involve the background research performed by either researchers or by AI. This can also involve whether the idea was proposed by researchers or by AI.

   Answer: **[D]**

   Explanation: The Humans provided some latest research papers on the Ising model and asked AI to come up with a project. AI did just that with a topic and all the hypotheses.

2. **Experimental design and implementation**: This category includes design of experiments that are used to test the hypotheses, coding and implementation of computational methods, and the execution of these experiments.

   Answer: **[C]**

   Explanation: The AI designs and implements the experiments under the supervision of the Human. The plan and code are all generated by AI, and the Humans check the results and provide feedback only in prompts to help AI obtain correct and compelling results. The feedback ranges from different levels, but the overall involvement of human labor is less than 20%

3. **Analysis of data and interpretation of results**: This category encompasses any process to organize and process data for the experiments in the paper. It also includes interpretations of the results of the study.

   Answer: **[D]**

   Explanation: Once the results are obtained, AI's performance in this regard is excellent, and the humans only need to tell AI to take action to analyze.

4. **Writing**: This includes any processes for compiling results, methods, etc., into the final paper form. This can involve not only writing of the main text but also figure-making, improving the layout of the manuscript, and formulation of the narrative.

   Answer: **[C]**

Explanation: AI carries out at least 95% of the text-writing and all of the figure-making. Humans are mainly responsible for providing high-level guidance. Humans also need to do some minor (for only a few words or a couple of sentences ) revisions when prompting is actually a layer of redundancy, or, even worse, a source of ambiguity, for such revisions. Some of the minor LaTeX adjustments are also made by humans to improve presentation quality (figure placement and layout).

5. **Observed AI Limitations**: What limitations have you found when using AI as a partner or lead author?

Description: AI does hallucinate and needs high-quality feedback/prompts. AI is also somewhat limited when it comes to compiling everything (code, results, figures) into a designated presentation format and style.

